# OpenReview forum: "PolygloToxicityPrompts: Multilingual Evaluation of Neural Toxic Degeneration in Large Language Models"
_colmweb.org/COLM/2024/Conference — COLM_

### Official Review · Reviewer_F15V · 2024-05-08

**Rating:** 7
**Confidence:** 2
**Ethics Flag:** 2

**Summary:**

The authors evaluated over 60 large language models (LMMs) with respect to their toxicity generation. They relied on a benchmark of 425K prompts that were evaluated for 17 languages. The toxity score was calculated with the PERSPECTIVE API tool. During the study, they evaluated the model size, language, among other attributes. The paper is well written and includes a good related work section.

**Ethics Concerns Details:**

Maybe due to the topic of the paper (toxicity language).

**Questions To Authors:**

- I wonder why just one industry tool was used for calculating the score and whether other tools could have been considered for comparison, even if just for some of the languages. How good does PERSPECTIVE API scores wrt. to other tools?

- I wonder whether any of the resources compiled in this work will be made available.

* minor comments:

- section 3.2: please define prompts and continuations.

- section 3.2: please explain GPT-4 tokens wrt. other tokens.

- Figure 3: please clarify why toxity is high if scores are under 0.5 (threshold defined in section 3.4).

- section 4.3: please define base, instruct, and preference, as well as the different preference-tuning methods (alignment).

**Reasons To Accept:**

- A comprehensive evaluation for a relativelly large number of prompts (425K), many languages (17), and many LLMs (60).
- Interesting insights on many aspects, e.g., alignment techniques, model categories, model size.

**Reasons To Reject:**

- It is unclear whether the selected tool (PERSPECTIVE API) is reliable for calculating the score.
- Other additional scoring tools could have been used for comparison, even if just for some of the languages.
- It is unclear whether any of the resources compiled in this work (dataset, prompts, scores, outputs) will be made available.

---

> ### Author Rebuttal · Authors · 2024-05-28
>
> Thank you for the feedback!
>
> ## Clarifying choice of toxicity detection tool
> We selected Perspective API since it is the industry standard tool for toxicity detection ([used by over 1000 partners](https://medium.com/jigsaw/announcing-experimental-bridging-attributes-in-perspective-api-578a9d59ac37)), is popularly used for toxicity studies in academia (e.g., [Gehman et. al. '20](https://aclanthology.org/2020.findings-emnlp.301.pdf),  [Si et. al. '22](https://dl.acm.org/doi/pdf/10.1145/3548606.3560599), [Hartvigsen et.al. '22](https://arxiv.org/pdf/2203.09509), [Lin et.al. '23](https://arxiv.org/abs/2310.17389)), and supports all languages that we have considered. Since toxicity detectors are classifiers trained on varying toxic data, we chose the most popular/commonly used detector available to us. Moreover, using Perspective API allows us to compare our dataset with existing works like [RealToxicityPrompts](https://aclanthology.org/2020.findings-emnlp.301.pdf), which used PerspectiveAPI as their default tool as well.
>
> **Reliability and accuracy of PerspectiveAPI:**
> Perspective API tabulates the area under the ROC curve (AUC-ROC) on various datasets, and achieves > 0.9 for almost all the supported languages, as shown in the [documentation](https://developers.perspectiveapi.com/s/about-the-api-model-cards?language=en_US&tabset-20254=3).
>
> **Using other tools for some languages only:**
> We have Llama Guard scores for the English subset for some models. We could compute toxicity scores using [Unitary’s Detoxify](https://github.com/unitaryai/detoxify), but the computation would not be complete within the discussion period. Moreover, during the dataset creation phase, we observed divergent scores between Detoxify and Perspective API (Pearson correlation between the scores was near 0). Hence, we proceeded with Perspective API, which is both an academic and industry-standard tool. Since there is public access to our dataset, we hope researchers will explore the use of other toxicity detection tools and compare them to Perspective API.
>
> ## Publicly available link to our resources
> We have made our dataset of prompts along with associated toxicity scores available in the shared repository (linked on page 1). We will also add code to generate continuations using LLMs to the repository.
>
> ## Minor comments for clarifications
> Thank you for the suggested minor clarifications, we will make the required changes to our manuscript.

---

> > ### Comment · Reviewer_F15V · 2024-06-04
> > **Thanks for the clarification**
> >
> > I thank the authors for clarifying the various points that I raised. I have no further questions.

---

### Official Review · Reviewer_pNqo · 2024-05-10

**Rating:** 8
**Confidence:** 4
**Ethics Flag:** 1

**Summary:**

The paper introduces PolygloToxicityPrompts, a dataset designed to assess toxicity generation across 17 languages. Using this dataset, authors evaluate 62 Language Model (LLM) variants, varying in both scale and architecture. Results show that model size inside the same family as well as preference tuning have an impact on the amount of toxicity generated.

**Questions To Authors:**

- What kind of toxicity is present in the dataset? It would be interesting to have more information about the specific phenomena in the dataset and how accurate are PERSPECTIVE API and LLama Guard on identify it.

**Reasons To Accept:**

- The study is exhaustive, including all of the more popular families of LLM models.
- The paper is well written and the process is easy to follow.
- The research questions are clear and show interesting patterns. Especially, the ones involving non-English results.

**Reasons To Reject:**

- Toxicity is a term difficult to define and evaluate. This paper uses third-party metrics to evaluate toxicity but it would be interesting to know better what aspects they are measuring in more detail. Toxicity could be evaluated as the generation of words that are considered toxic (e.g. HolisticBias (https://arxiv.org/abs/2305.13198)), but that would ignore toxic examples that do not include toxic words out of context. (e.g. "I think this reviewer is not intelligent", not of the words could be considered toxic if isolated).

---

> ### Author Rebuttal · Authors · 2024-05-28
>
> Thank you for the positive and thoughtful feedback! We agree that the scope of measuring toxicity is very open-ended. Hence, we are using Jigsaw’s definition of toxicity as measured by Perspective API. In addition to the TOXICITY score, Perspective API also provides scores for IDENTITY ATTACK, INSULT, PROFANITY, SEVERE_TOXICITY, and THREAT as stated in their [documentation](https://developers.perspectiveapi.com/s/about-the-api-attributes-and-languages?language=en_US ).
>
> We will provide additional data analysis based on the attributes mentioned above in our manuscript, to explore what kinds of toxicity are generated by different models. Moreover, we will provide the Llama Guard scores for the English subset of our dataset along with the distribution of harmful content in the subset as per Llama Guard’s taxonomy.

---

### Official Review · Reviewer_WPkF · 2024-05-12

**Rating:** 9
**Confidence:** 4
**Ethics Flag:** 1

**Summary:**

The paper introduces a new dataset, PolygloToxicityPrompts, consisting of 245K prompts in 17 languages with varying toxicity levels. It is inspired by RealToxicityPrompts and includes prompts in the 17 languages supported by Perspective API, which supplies toxicity ratings. This language set covers a range of language families while excluding low-resource languages.

The paper evaluates 62 LLMs including base, instruction tuned, and preference tuned models. The experiments investigate the effects of per-language training data size, model size, alignment methods, and instruction following ability. (Experiments are run on a  subset of 5K prompts per language due to the large number of models investigated.)

**Questions To Authors:**

Given the source of the prompts in publicly available corpora, what role does memorization play in toxic outputs?

**Reasons To Accept:**

- Clearly written, well motivated, and well supported with prior work. Clear limitations and ethids statements.
- A large benchmark of significant use to the field.
- Benchmark is 83% natural text and not a translated version of English benchmarks.
- Large number of LLMs evaluated.
- Evaluation metrics are clearly explained and appropriate.
- Identifies some subtle confounders such as models that produce shorter or poor quality responses appearing to have lower toxicity.

**Reasons To Reject:**

I don't see any reasons to reject.

---

> ### Author Rebuttal · Authors · 2024-05-28
>
> Thank you for the encouraging feedback and positive endorsement of our work!
>
> To address your question: since the training data for nearly all models considered in our study are proprietary, it’s hard to make a comment on the role memorization would play in toxic outputs. However, we sample 10 generations per prompt with a high temperature setting to introduce randomization and alleviate concerns with memorization. We hope that more LLMs will be released along with their pretraining data (as is the plan for AI2's OLMo), which will enable more direct studies of this phenomenon that until now only companies themselves have been able to do (e.g., [Longpre et al '23](http://arxiv.org/abs/2305.13169)).

---

> > ### Comment · Reviewer_WPkF · 2024-06-04
> >
> > Thanks for the reply. The explanation of multiple samples at high temperature to minimize memorization effects sounds like a reasonable approach.

---

### Official Review · Reviewer_sRPx · 2024-05-19

**Rating:** 10
**Confidence:** 4
**Ethics Flag:** 1

**Summary:**

This paper provides a multilingual dataset for eliciting toxic responses in large language models. The authors systematically collected an extensive collection of *25000 prompts* per language covering *17 languages*. Furthermore, using the dataset, this paper provides a detailed analysis of toxicity in large language models by considering variables such as prompt language, model size, model alignment methods, and toxicity evaluation methods.

**Reasons To Accept:**

* The key contributions of this paper are: a) a large multilingual corpus for evaluating toxicity, and b) analyzing current open and closed source models based on these prompts
* The resources provided in this paper would enable future research on toxicity in multilingual generations
* The paper also presents novel insights into studying toxic generation. For example, instruction tuning and preference tuning play a role in reducing toxicity. Interestingly, specific alignment methods such as DPO or SFT-PPO do not lead to significant changes in toxicity.
* Last but not least, this paper is very well written, easy to follow, and provides clear reproducibility statements indirectly enabling further research in developing similar benchmarks for LLMs.

**Reasons To Reject:**

N/A

---

> ### Author Rebuttal · Authors · 2024-05-28
>
> Thank you for the encouraging feedback and the very positive review!

---

> > ### Comment · Reviewer_sRPx · 2024-06-04
> >
> > Great, once again, thank you for the good work!

---

### Official Review · Reviewer_JsRK · 2024-05-20

**Rating:** 7
**Confidence:** 3
**Ethics Flag:** 1

**Summary:**

The authors present POLYGLOTOXICITYPROMPTS (PTP). A multilingual toxicity
evaluation benchmark which consists of 425K prompts spanning 17
languages. Further, the authors use the PTP benchmark to investigate research questions to study the impact of model size, prompt language, and instruction and preference-tuning methods on toxicity by benchmarking over 60 LLMs.

**Questions To Authors:**

Refer above.

1. The README in the shared repository is not very helpful. It's just an abstract.

**Reasons To Accept:**

+ The authors do a thorough job in including diverse languages in building the toxicity dataset from C4 and Pile.
+ The authors present their evaluation a wide number of LLM including open-source LLMs.
+ The authors present insights which can be ripe for future work, for example, the correlation between input and output toxicity for different models.

**Reasons To Reject:**

The authors clarify that to attain a larger sample of toxic content for languages with low toxicity rates, they create
synthetic high-toxicity data. Specifically, they translate toxic samples from the mC4 and THE
PILE corpora into target languages using the NLLB-3B model to create ≈ 70K translated prompts across 9 languages. While the fact that this  amounts to only 16.8% of our dataset would have been a non-issue, and in-fact heralded in other contexts.. given that the goal of this paper is to be multilingual, it makes me wonder if this is incorrect in principle? While, I don't think this impacts the value of the rest of the dataset, the authors should make more explicit right upfront, or elsewhere!

---

> ### Author Rebuttal · Authors · 2024-05-28
>
> Thank you for the feedback! While we do have a subsection dedicated detailing these challenges and our process for creating synthetic high-toxicity data, we will revise the introduction to make it more explicit that our dataset contains some amount of synthetically generated translated data as well.
>
> We will also update the shared repository with code to load our dataset and generate continuations with LLMs, as well as a more detailed README.

---

> > ### Comment · Reviewer_JsRK · 2024-06-05
> >
> > Thank you!
> > With regards to the shared repository, the README is still not updated? I'm slightly confused on the utility and motive of sharing an anonymous github repository without a bare minimum README to navigate it? Right now, all it has is the abstract from the paper.

---

> > > ### Author Response · Authors · 2024-06-07
> > >
> > > Thank you for the follow up! We apologize for the delay in updating the README, we have now updated it with details about the dataset. Due to our dataset's size and Github's storage constraints, we are currently unable to add code to the repository. We plan to move our dataset to huggingface for easier access and update the repository with code to evaluate toxic degenerations in arbitrary LLMs after the discussion period to maintain anonymity. We hope that you find the README more helpful now.

---

### Decision · Program_Chairs · 2024-07-10

**Decision:**

Accept

**Comment:**

The paper design and curate a large multilingual benchmark (245k) to measure toxicity across 17 languages; This is a valuable resource for future work, especially given that the multilingual prompts---for the most part---are not translation from English. The authors have evaluated a large number of LLMs (62) on this benchmark, providing insights on the role of various factors such as model size, etc on toxicity generation. I agree with the reviewers about the impact of this work, and encourage the authors to apply reviewers' comments. Another interesting question to explore is whether the performance of models on natural prompts correlate with the synthetically generated ones (and whether the latter can be useful replacement).